# Aging Stability Analysis of Slope Considering Cumulative Effect of Freeze–Thaw Damage—A Case Study

Zhiguo Chang [1,2,*], Weiguang Zhang [2,3,*], Gang Zhao [4], Fa Dong [3] and Xinyu Geng [3]

1 School of Safety Science and Engineering, Xinjiang Institute of Engineering, Urumqi 830000, China
2 Xinjiang Key Laboratory of Geohazard Prevention, Urumqi 830000, China
3 School of Mining Engineering and Geology, Xinjiang Institute of Engineering, Urumqi 830000, China; df380222172@163.com (F.D.); ts19020096p21@cumt.edu.cn (X.G.)
4 Team of Comprehensive Geological Exploration, Coalfield Geology Bureau of Xinjiang Uygur Autonomous Region, Urumqi 830000, China; gangzhao1314@163.com
* Correspondence: changzg@cumt.edu.cn (Z.C.); weiguang1228@126.com (W.Z.)

**Abstract:** The change of physical and mechanical properties of slope rock mass in open-pit mines in seasonally frozen area under the action of freeze–thaw cycles is one of the main reasons for slope instability. In this paper, taking the mechanical parameters of coal seam and sandstone layer in the Beitashan Pasture Open-Pit Mine in Xinjiang as the research object, considering the combined effect of the frost-heave tensile stress in the crack perpendicular to the crack surface and the three-dimensional confining pressure in the crack, the criterion for cracking of fractured rock mass under freeze-thaw condition is determined by applying the principle of stress superposition and the theory of strain energy density factor, and the theoretical frost-heave stress required for cracking is deduced. On this basis, the sensitivity analysis of the fixed factors and variable factors to the theoretical frost-heave stress was performed, respectively. Finite element analysis was utilized to analyze the slope stability under the attenuation of five groups of different rock mass mechanical properties and to determine the slope angle required for the slope stability. Seven different slope angles of sidewall mining ranging from 36° to 51° are analyzed. The results of finite element analysis show that considering the timeliness difference of rock mass parameters with time, the safety factor of slope is reduced from the original 1.70 to 1.18, and 91,500 tons of coal resources can be recovered every year, with remarkable economic benefits.

**Keywords:** open-pit mine; slope stability; cumulative damage effect; strength reduction method

## 1. Introduction

From the excavation of the open-pit mine to the final reclamation, there are five mining unit operations, including drilling, blasting, excavation, transportation, and dumping. The disturbance and damage to the original stratum in this process is an irreversible process that lasts for a long time [1]. The main mining methods of open-pit mines are shown in Table 1. It can be seen that whether the open-pit dump is set outside the pit or inside the pit, it takes a long time from mining to backfilling. As the slope is exposed to the natural environment and affected by temperature, seepage, mining, freezing, and thawing [2,3], its structural integrity and rock strength will gradually decrease as the time of slope exposure increases [4,5]. The change of the physical and mechanical properties directly affects the long-term sustainable stability of the slope engineering [6].

The distribution of seasonal frozen soil in China extends from the north of the Yangtze River to north, northwest, and northeast China, accounting for about half of the land area [7]. China's open-pit coal mines are mainly distributed in Inner Mongolia, Xinjiang, Shanxi, and Yunnan and other border areas, which are mainly seasonally frozen soil areas except for Yunnan Province [8]. Differences in latitude and altitude will lead to huge differences in freezing depth. In temperate regions, the freezing depth is below 1 m; in

cold areas, it can reach about 3 m; and in some alpine regions, it can even exceed 5 m [9]. There are huge differences in physical and mechanical properties of soil in freezing and melting states [10,11]. Roman's study [12] on the long-term frost resistance of silty sand shows that frozen soil has high strength, high hardness, and small deformation. After the frozen soil melts, the ice connecting the soil particles melts into water, which will lead to the increase of soil plasticity and the sharp decrease of strength. If the water is sufficient, it even exhibits strong rheological characteristics [13,14].

**Table 1.** Mining methods applied on coal mine excavation-backfilling process.

| Mining Method | Equipment Application | External Dump | Internal Dump |
|---|---|---|---|
| Non-continuous mining system | Single-bucket excavator + Truck + Bulldozer Dragline + Bulldozer | The only choice in the initial stage of mining, resulting in long exposure time of slope | Short haul distance and low cost but only applicable to horizontal or near-horizontal strata |
| Continuous mining system | Bucket-wheel excavator + Belt conveyor + Stacker + Bulldozer | | |
| Semi-continuous mining system | Single-bucket excavator + Truck + Fixed crushing station + Belt conveyor + Stacker + Bulldozer Single-bucket excavator + Mobile crusher + Belt conveyor + Stacker + Bulldozer | | |

Not only is the topsoil greatly affected by freezing and thawing, but the rock mass is also sensitive to freezing and thawing damage [15]. During the formation of rock mass, microstructure and microdefects of different types, scales, and occurrence will be formed [16,17]. The mutual development, connection, and expansion of such structures and defects will affect the deformation and failure of rock mass [18,19]. For the geotechnical engineering in the seasonally frozen area, the sufficient fissure water in the rock slope freezes in winter and thaws in the spring or summer in the following year, forming a strong freeze–thaw cycle in years [20,21]. Meanwhile, the temperature difference between day and night can also cause a weak freeze–thaw cycle in a day in spring and autumn [22]. Yavuz [23] conducted freeze–thaw cycle tests under saturated water conditions on three kings of rock samples of marble, limestone, and tuff. The freeze–thaw cycle causes the phase change of water. When the water freezes into ice, due to the action of ice wedge pressure, it leads to the expansion of cracks and pores in the rock mass, resulting in the fragmentation and loosening of the rock mass. Wang [24] studied the meso-damage and permeability evolution characteristics of rocks under freeze–thaw cycles by nuclear magnetic resonance. With the increase of cycle times, the pore structure of rock samples changed with the expansion of pore size and the increase of pore number, and the average porosity increased accordingly. Takarli [25] conducted freeze–thaw cycle tests on granite under saturated and unsaturated conditions and found that when the saturation of granite reached 70%, the fissure rate and the fracture growth rate were 1.7 times and 5 times that of granite with 30% saturation after 50 freeze–thaw cycle tests. Crack generation and propagation in rock is intricate not only in the elusory fracturing network, but the mechanical behavior of materials may also be altered [26]. With the development of science and technology in recent years, modern detection methods [27,28], numerical simulation methods [29,30], and high-end experimental equipment [31] have been applied to the research of this kind of problems. The development process of the internal defect structure of rock mass is formed slowly with the passage of time. In the entire evolution process of geotechnical engineering, change in external environmental factors may lead to the gradual deterioration of its mechanical properties [32]. The time-dependent characteristic of the gradual weakening of the internal structural plane of rock mass with time is very important for the slope from microscopic damage to macroscopic instability [33].

Accurate safety and stability evaluation of the slope is an important guarantee to ensure the sustainable production of engineering construction [34,35]. There are many methods for slope stability analysis, such as limit equilibrium method [36], finite element method, discrete element method, boundary element method, etc. During the evaluation process, the selection of system evaluation parameters directly affects the final output results. In the traditional slope stability analysis, the slope is usually regarded as a static

object. In the early stage of design, the basic mechanical parameters of soil and rock mass in various layers of the slope are obtained through engineering analogy method or physical and mechanical experiment of rock blocks in the laboratory, and the safety factor of the slope is determined by means of a computer using a specific stability analysis method. The traditional design method ignores the influence of time effect on geotechnical mechanical parameters of slope. The purpose of this study is to present the sensitivity of the freeze–thaw damage effect to the physical and mechanical properties of the slope and to study the aging stability of the slope under the condition of rock and soil deterioration for giving the safe construction design and protection suggestions for aging slopes in open-pit mines.

## 2. Mechanism

During the diagenesis of rock mass, cracks with different shapes are formed under the action of geological movement, external power, and in situ stress. The internal pores and micro-cracks of rock formations located in seasonally frozen soil areas with abundant water sources are often filled with water. When the ambient temperature drops below the freezing point in cold winter or at night, the water gradually freezes and expands. It may cause the surrounding rock cracks to crack and extend when the expansion stress of the ice exceeds the cracking strength of the rock cracks. When the temperature rises above the freezing point, new water is added to continue to expand the micro-crack space after the ice melts. The freezing and melting process caused by the periodic change of temperature occurs alternately and repeatedly, resulting in the continuous accumulation of damage and deterioration of ground soil and rock layer and the gradual reduction of mechanical strength. The freeze–thaw cracking process of fractured rock mass is shown in Figure 1.

Most of the preexisting cracks in rock masses are subjected to complex loading conditions [37]. For saturated rock mass with cracks of different dip angles, the stress during frost-heave can be divided into: (1) frost-heave tensile stress in the crack perpendicular to the crack surface and (2) three-dimensional confining pressure.

Consider a series of equal-width and straight-line collinear cracks oriented at a crack inclination angle $\beta$ and propagating at an angle $\theta$ inside the rock. Setting the origin at the midpoint of any crack, a local x-y coordinates system with one axis along the crack is stabled as shown in Figure 2. Assume the crack length of $2a$, the center distance of the adjacent cracks of $2b$, the vertical stress as $\sigma = \gamma H$, the horizontal stress as $\lambda\sigma$, and the frost-heave tensile stress perpendicular to the crack wall caused by water phase change as $\sigma_T$.

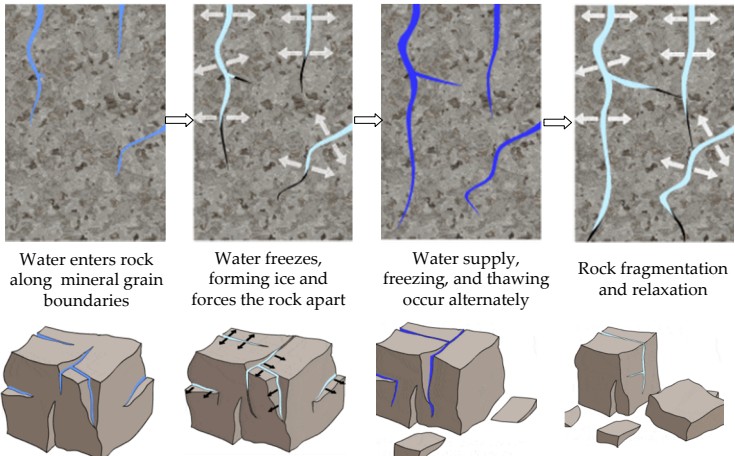

| Water enters rock along mineral grain boundaries | Water freezes, forming ice and forces the rock apart | Water supply, freezing, and thawing occur alternately | Rock fragmentation and relaxation |

**Figure 1.** Freeze–thaw cracking process of fractured rock mass.

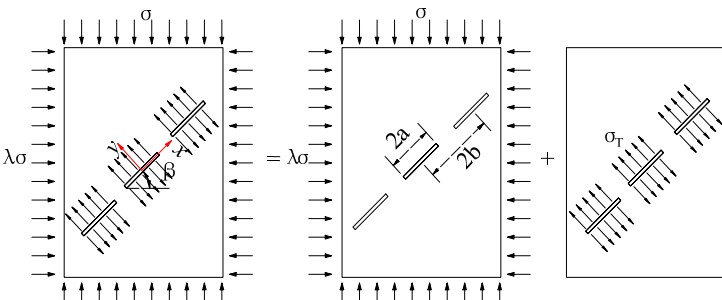

**Figure 2.** Stress model of rock mass after fissure water freezing.

Then, the normal stress component $\sigma_\beta$ and tangential shear stress component $\tau_\beta$ acting on the crack surface are obtained, respectively,

$$
\begin{aligned}
\sigma_\beta &= \sigma_T - \left( \frac{\sigma_1 + \sigma_3}{2} + \frac{\sigma_1 - \sigma_3}{2} \cos 2\beta \right) = \sigma_T - \gamma H (\lambda \cos^2 \beta + \sin^2 \beta) \\
\tau_\beta &= \frac{\sigma_1 - \sigma_3}{2} \sin 2\beta = \gamma H (\lambda - 1) \sin \beta \cos \beta
\end{aligned}
\tag{1}
$$

Due to the repeated freeze–thaw cycle and confining pressure, the opening–sliding failure occurs under the combined action of normal stress $\sigma_\beta$ and shear stress $\tau_\beta$, which can be summarized as compression shear composite crack [38], i.e., Mixed Mode I–II crack (Mode I–II for short). According to the principle of stress superposition, the stress intensity factor under complex loads is equal to the sum of the stress intensity factors of each individual load. Then the stress distribution at the tip of the Mode I–II shown in Figure 2 can be obtained by superimposing the solution of Mode I crack and Mode II crack. Hence, the stress components of cracks tip in the polar coordinates for the Mode I–II are given as

$$
\left.
\begin{aligned}
\sigma_x &= \frac{1}{\sqrt{2\pi r}} \left[ K_I \cos \frac{\theta}{2} \left( 1 - \sin \frac{\theta}{2} \sin \frac{3\theta}{2} \right) - K_{II} \sin \frac{\theta}{2} \left( 2 + \cos \frac{\theta}{2} \cos \frac{3\theta}{2} \right) \right] \\
\sigma_y &= \frac{1}{\sqrt{2\pi r}} \left[ K_I \cos \frac{\theta}{2} \left( 1 + \sin \frac{\theta}{2} \sin \frac{3\theta}{2} \right) + K_{II} \sin \frac{\theta}{2} \cos \frac{\theta}{2} \cos \frac{3\theta}{2} \right] \\
\tau_{xy} &= \frac{1}{\sqrt{2\pi r}} \left[ K_I \sin \frac{\theta}{2} \cos \frac{\theta}{2} \cos \frac{3\theta}{2} + K_{II} \cos \frac{\theta}{2} \left( 1 - \sin \frac{\theta}{2} \sin \frac{3\theta}{2} \right) \right]
\end{aligned}
\right\}
\tag{2}
$$

where $K_I$, $K_{II}$ are the stress intensity factors in Mode I and Mode II, respectively, which are defined as

$$
K_I = \sigma_\beta \sqrt{\pi a} \left( \frac{2b}{\pi a} \tan \frac{\pi a}{2b} \right)^{\frac{1}{2}}
\tag{3}
$$

$$
K_{II} = \tau_\beta \sqrt{\pi a} \left( \frac{2b}{\pi a} \tan \frac{\pi a}{2b} \right)^{\frac{1}{2}}
\tag{4}
$$

The strain energy density factor at the tip of Mode I–II is expressed by

$$
S = a_{11} K_I^2 + 2 a_{12} K_I K_{II} + a_{22} K_{II}^2
\tag{5}
$$

where

$$
\left.
\begin{aligned}
a_{11} &= \frac{1}{16\pi G} [(3 - 4\mu - \cos \theta)(1 + \cos \theta)] \\
a_{12} &= \frac{1}{16\pi G} (2 \sin \theta)[\cos \theta - (1 - 2\mu)] \\
a_{22} &= \frac{1}{16\pi G} [4(1 - \mu)(1 - \cos \theta) + (1 + \cos \theta)(3 \cos \theta - 1)]
\end{aligned}
\right\}
\tag{6}
$$

where $G$ is shear modulus, and $\mu$ is Poisson's ratio.

Substituting Equations (1), (3), and (4) into Equation (5) leads to

$$
S = 2b \tan \frac{\pi a}{2b} \cdot F(\beta, \theta)
\tag{7}
$$

where $F(\beta, \theta)$ is

$$
\begin{aligned}
F(\beta, \theta) = \ & a_{11} \left[ \sigma_T - \gamma H \left( \lambda \cos^2 \beta + \sin^2 \beta \right) \right]^2 \\
& + a_{12} \left[ \sigma_T - \gamma H \left( \lambda \cos^2 \beta + \sin^2 \beta \right) \right] (\lambda - 1) \sin 2\beta + a_{22} (\lambda - 1)^2 \sin^2 \beta \cos^2 \beta
\end{aligned}
$$

According to the theory of strain-energy-density factor, the crack will propagate along the direction $\theta_0$ with the minimum strain-energy-density factor when the strain-energy-density factor $S$ reaches the critical value $S_C$; that is,

$$
\frac{\partial S}{\partial \theta} = 0, \ \frac{\partial^2 S}{\partial \theta^2} > 0, \ S_{\min} = S_C \tag{8}
$$

Sih [39] analyzed the relationship between the cracking angle and the crack inclination angle when different Poisson's ratios are used as shown in Figure 3. Obviously, the crack propagation direction of the Mode I–II is approximately perpendicular to the direction of normal stress and satisfies $\beta + |\theta_0| \rightarrow 90°$.

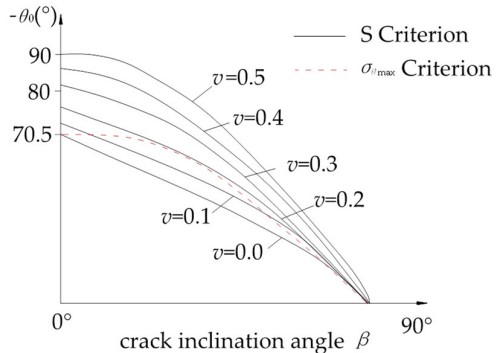

**Figure 3.** Relationship between the cracking angle and the crack inclination angle corresponding to different Poisson's ratio.

It can be seen that for a given crack inclination angle $\beta$, the value of the cracking angle $\theta_0$ can be obtained according to Formula (8), and it can be taken into Formula (7) to obtain the cracking criterion of the freezing-thawing fractured rock mass:

$$
S_{\min} = 2b \tan \frac{\pi a}{2b} F(\beta, \theta_0) = S_C \tag{9}
$$

where $S_C$ is a characteristic index indicating the cracking strength of rock mass, which can be determined by the crack toughness value $K_{IC}$ of the Mode I, i.e,

$$
S_C = \frac{1 - 2\mu}{4\pi G} K_{IC}^2 \tag{10}
$$

By substituting Equation (10) into Equation (9), the theoretical frost-heave stress required for freeze–thaw crack cracking can be calculated as follows:

$$
\sigma_T = \frac{A a'_{11} - B a'_{12} + \sqrt{B^2 a'^{2}_{12} - B^2 a'_{11} a'_{22} + \frac{1 - 2\mu}{4\pi^2 G b \tan \frac{\pi a}{2b}} K_{IC}^2 a'_{11}}}{a'_{11}} \tag{11}
$$

where

$a'_{11} = \frac{1}{16\pi G} [(3 - 4\mu - \cos \theta_0)(1 + \cos \theta_0)]$,

$a'_{12} = \frac{1}{16\pi G} (2 \sin \theta_0)[\cos \theta_0 - (1 - 2\mu)]$,

$a'_{22} = \frac{1}{16\pi G} [4(1 - \mu)(1 - \cos \theta_0) + (1 + \cos \theta_0)(3 \cos \theta_0 - 1)]$,

$A = \gamma H (\lambda \cos^2 \beta + \sin^2 \beta)$,

$B = \gamma H (\lambda - 1) \sin \beta \cos \beta$.

## 3. Results

Based on the stress analysis of rock mass with tension shear composite cracks, the analytic relation in theoretical frost-heave stress $\sigma_T$ required for crack cracking of high and steep composite slope under freeze–thaw conditions is built up. The factors affecting the theoretical frost-heave stress mainly includes the crack initial length $2a$, adjacent crack distance $2b$, crack inclination angle $\beta$, burial depth $H$, shear modulus G, Poisson's ratio $\mu$, and fracture toughness $K_{IC}$. Among them, fracture toughness $K_{IC}$ describes the ability of rock to prevent crack propagation, just like Poisson's ratio $\mu$ and shear modulus $G$; both are inherent characteristics of rock materials, which can be summarized as fixed factors independent of the size and shape of the crack itself, while the variable factors, such as crack initial length $2a$, adjacent crack distance $2b$, and crack inclination angle $\beta$, are the parameters that describe the frequency and shape of the crack distribution, whose approximate value range can be determined by distribution in the rock. Therefore, the theoretical frost-heave stress $\sigma_T$ required for crack freeze–thaw cracking can be determined according to the crack distribution law of rock mass with different lithology.

In this section, taking the coal seam and sandstone layer above it in the Beitashan Pasture Open-Pit Mine (BP Mine) in Xinjiang as a prototype, the sensitivity of relevant parameters to theoretical frost-heave force is discussed. According to the field measured data, the mining depth at the end slope of the open pit is 80 m, the average thickness of the coal seam is 23 m, and the thickness of the sandstone at the roof of the coal seam is 29 m. The selection of mechanical parameters and fracture parameters is shown in Table 2.

**Table 2.** Mechanical parameters of rock mass.

| Stratum | Young's Modulus (GPa) | Poisson's Ratio | Fracture Toughness (MPa·m$^{1/2}$) | Crack Length (m) | Crack Ratio |
|---|---|---|---|---|---|
| Sandstone | 3.12 | 0.214 | 28.6 | 0.02–0.3 | <15 |
| Coal | 0.24 | 0.36 | 13.0 | 0.02–0.3 | <15 |

Firstly, sandstone is taken as the research object. The theoretical frost-heave force under different buried depths is shown in Figure 4 by incorporating the parameters in Table 2 into Equation (11). The curves show that the variation of limit height with the crack dip angle is consistent, showing a significant trend of decreasing first and then increasing. The curve shows that when other parameters are fixed, the theoretical frost-heave force increases with the increase of the mining depth, showing a linear growth relationship. That is, with the increase of the mining depth, the vertical pressure generated by the overlying rock layer gradually increases, producing vertical positive pressure on the cracks in the rock mass along the strike direction. If the frost-heave fractures occur, the theoretical frost-heave force need to be increased accordingly.

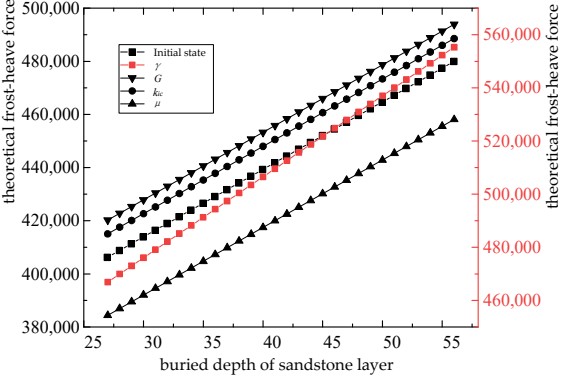

**Figure 4.** Sensitivity analysis of fixed parameters with buried depth.

However, the frost-heave force is caused by the change of water phase state and volume expansion in the crack. The maximum volume can be increased by 9% after water is transformed into ice. Therefore, the actual frost-heave force should gradually increase until it tends to be stable. The actual frost-heave force is usually lower than the theoretical frost-heave force in case of crack cracking. In addition, the other four straight lines in the Figure 4 respectively represent the variation trend of theoretical frost-heave force with the increase of mining depth when the shear modulus, average weight of overlying rock mass, Poisson's ratio and fracture toughness of fractured rock mass increase by 20%. It can be seen that when the shear modulus, Poisson's ratio and fracture toughness of fractured rock mass increase, the theoretical frost-heave force only changes numerically and remains the same in the growth rate, that is, the slope of the straight line section remains unchanged. The increase of the average gravity of the overlying rock mass leads to the increase of the initial value and growth rate of the theoretical frost-heave force. Therefore, under the same mining conditions, the deeper the mining depth and the greater the load of the overlying strata, the less prone the rock mass to frost-heave cracking.

Figure 5 shows the relationship between the mining depth and the frost-heave force required for frost-heave cracking of coal seam and sandstone layer under the same mining conditions. It can be seen that although the coal seam is excavated deeper than the rock layer, the frost-heave force required for coal cracking is lower than that of sandstone. This phenomenon is caused by different strength of rock mass. Combined with Figure 4, it can be seen that rocks with small shear modulus, small fracture toughness and large Poisson's ratio are more prone to frost-heave fracture failure.

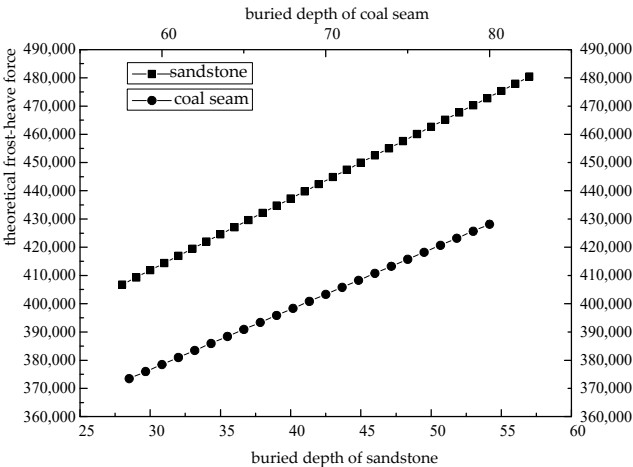

**Figure 5.** Relationship between frost heaving force required for cracking and mining depth.

The variable factors like the crack length, crack spacing and crack inclination angle are important parameters to characterize the distribution frequency and shape of cracks, whose distribution shape and density directly affect the strength, permeability and freeze-thaw weathering resistance of engineering rock mass. The approximate interval of the frequency and shape parameters of the cracks in the coal seam was obtained by borehole peeper for BP Mine. The results of borehole observation show that the mine has simple stratum structure under small tectonic movement. Cracks are mainly distributed near the entrance of the borehole, in the way of high frequency and short spacing cluster with long spacing cracks scattered randomly and irregularly among clusters. According to the spatial distribution law of internal cracks in the slope, the initial crack length is set to be 2 cm–30 cm, the crack ratio is 1–15, and the crack inclination angle is 0–90° to analyze the influence of different crack densities and shapes on the theoretical frost-heave force. Figure 6 shows the relationship between the theoretical frost-heave force and the crack inclination angle at different mining depths when the initial length and spacing of the cracks are constant. As the crack inclination angle changes from horizontal to vertical, the theoretical frost-heave

force gradually decreases until it becomes stable, and the rate of change is smaller in the near-horizontal and near-vertical directions, and greater in the inclined direction at 20–70°. The vertical cracks are more prone to frost-heave fracture than horizontal cracks at the same mining depth. Combined with the mechanical model shown in Figure 2, the decomposition force of the overlying rock mass in the direction perpendicular to the cracks gradually decreases with higher degree of the inclination angle of the cracks, which is more likely to cause the cracks to crack and expand under the action of frost heave. In addition, the six curves of the two rock masses at different excavation depths shows that the theoretical frost-heave force of the same lithological rock mass changes with the inclination angle of the crack in the same way, while the change range of coal is larger than that of sandstone, indicating that the cracks in the coal seam are more sensitive to frost heave.

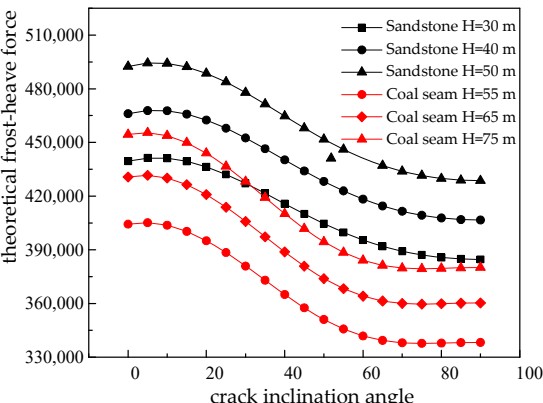

**Figure 6.** Relationship between theoretical frost heaving force and crack inclination angle.

The crack ratio is an important indicator reflecting the degree of crack development in geological engineering, which can be expressed by the ratio of the average distance between adjacent cracks to the length, that is, b:a. The single curve in Figure 7 shows that the theoretical frost-heave force is logarithmically increased with the crack ratio, which means when the crack length is constant, the more dispersed the cracks are, and the fewer frost-heave fractures occurs between adjacent cracks. Specifically, the gradient change of theoretical frost-heave force is most significant when the crack ratio is between 1–2. In this case, the distance between adjacent cracks is relatively smaller to the length of the crack itself, and insufficient strength to effectively resist fracture is easy to cause penetration failure between cracks; when the crack ratio is between 2–4, the variation of theoretical frost-heave force gradually decreases. The main reason is that the elastic core area in the rock mass between adjacent cracks gradually increases, which is not easy to fracture failure; when the crack ratio > 4, the curve gradually tends to be horizontal, and the theoretical frost-heave force approaches a fixed value and tends to be stable; that is, the frost-heave damage between adjacent cracks can be regarded as an independent process, with almost no influence on each other.

It can also be seen from Figure 7 that when the crack ratio is between 1–2, the theoretical frost-heave force varies greatly with the crack length. With the increase of the crack length, the curve slope of the theoretical frost-heave force increases with the crack ratio. Therefore, when the change trend of theoretical frost-heave force tends to balance, the greater the crack length, the greater the required theoretical frost-heave force; thus, fractured rock mass is less prone to frost-heave fracture. This explains the entire frost-heave failure process of the fractured rock mass. For saturated micro cracks, the theoretical frost-heave force required for the micro crack length is small. When the frost-heave force generated by the water phase transition in the crack is greater than the theoretical frost-heave force, stress concentration will occur at the end of the crack, prompting cracks to grow along the crack direction, which may cause the original crack length be expanded. As the crack length expands, the theoretical frost-heave force required for fracture also increases

and gradually approaches the actual frost-heave force, so the frost-heave damage range gradually stabilizes. Then, when the actual frost-heave force does not meet the ultimate tensile stress of rock mass cracking, the crack will not directly propagate, but a progressive damage zone will be formed at the crack tip. Damage accumulation to a certain extent will lead to the initiation and propagation of new cracks. Therefore, the freeze-thaw damage process can be regarded as a fatigue damage accumulation process, which well explains the reason why the degradation rate of physical and mechanical properties of the sample decreases or even tends to be stable after repeated freeze–thaw cycle experiments.

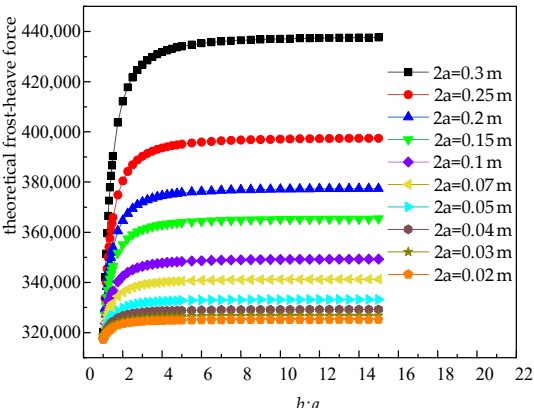

**Figure 7.** Relationship between theoretical frost heaving force and crack ratio.

## 4. Discussion

It takes a long time from excavation to formation for the slope of open-pit mine, especially for large, deep open-pit mines. In the process of mining step by step, the mine must advance horizontally for a sufficient distance before it can continue to the descending section. The deeper the open-pit excavation is, the greater the difference in the exposure time of the rock and soil between the upper and lower parts of the slope in the vertical space, resulting in different deterioration degrees of rock and soil strength [40]. The upper steps are exposed early and are greatly affected by engineering and the environment, which are characterized by large strength attenuation and low stability, while the lower steps are exposed late, with small impact, appearing as weak strength attenuation and high stability. The upper and lower parts of the slope show obvious spatial aging differences [41] as shown in Figure 8.

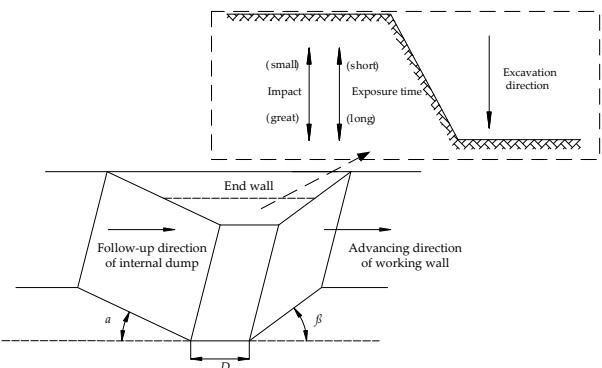

**Figure 8.** Timeliness difference of mechanical properties of slope in space.

The distance *D* between the bottom line of the lowest bench of the working wall and internal dump is the distance with the shortest exposure time of the entire end wall. Whether the internal dump can be followed up in a timely manner affects the reserved distance at the bottom of the pit, which further affects the overall exposure time of the end

wall on both sides. In traditional open-pit mine design, the safety factor of the slope is set as a fixed value, ignoring the external influence on the rock mass of the exposed slope. Therefore, this paper proposes an aging slope theory that considers the influence of time effect on the properties of rock and soil of slope.

Taking the engineering geological conditions of the BP Mine as an example, an idealized horizontal occurrence geological model was established after simplification. Next, the traditional slope and the slope considering the damage effect will be modeled and calculated, and then, the corresponding relationship between the aging slope angle and time can be determined according to the comparison of the stability coefficient between the traditional slope and the slope considering the damage effect. The modeling flowchart is shown in Figure 9.

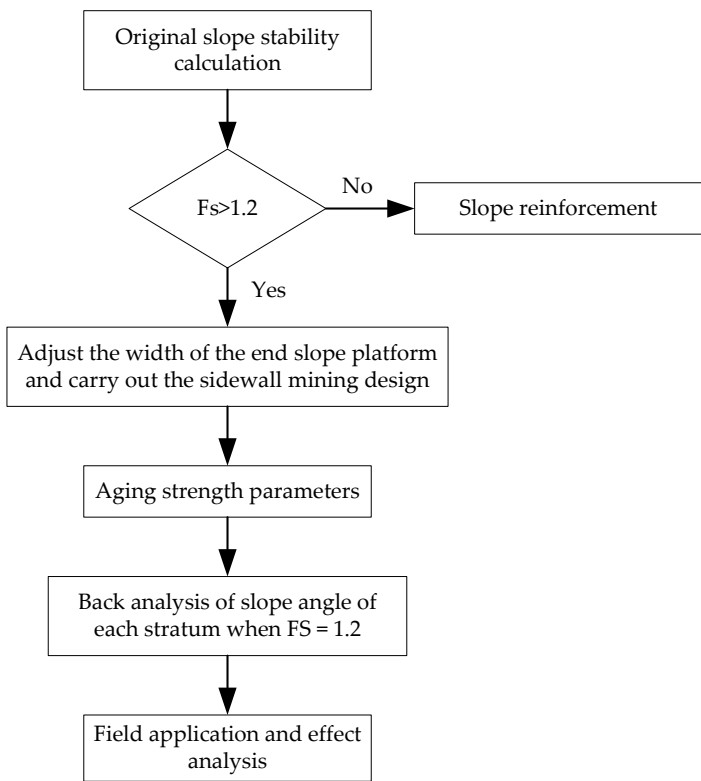

**Figure 9.** Design flowchart of aging slope.

The FLAC3D 3.0 was used to perform the initial balance calculation of the model, and the safety factor is 1.70, which is too conservative, causing a large amount of coal resources to be covered under the edge. This virtually increases the production stripping ratio of the mine, resulting in a waste of resources. The models with the final slope angle values of 36°, 38°, 40°, 42°, 45°, 48°, and 51° are established by reducing 1 m slope step width each time starting from 14 m. Taking the friction angle, cohesion, and the gravity of the rock mass as the freeze–thaw attenuation factors, five groups of tests are designed to calculate the safety factors of the slope under the attenuation of mechanical parameters as shown in Table 3.

**Table 3.** Reduction of aging parameters in each group.

| Aging Parameters | Group 1 | Group 2 | Group 3 | Group 4 | Group 5 |
|---|---|---|---|---|---|
| Cohesion | −5% | −10% | −15% | −20% | ±0% |
| Friction angle | −5% | −10% | −15% | −20% | ±0% |
| Gravity | +5% | +10% | +15% | +20% | ±0% |

Take the first group of reductions as an example. Seven different slope angles are designed by adjusting the width of end slope. When the aging parameters of the slope are degraded by 5%, the factors of safety (Fs for short) are 1.62, 1.48, 1.35, 1.23, 1.12, 1.04, and 0.96 when the slope angle is 36°, 38°, 40°, 42°, 45°, 48°, and 51°, respectively, as shown in Figure 10.

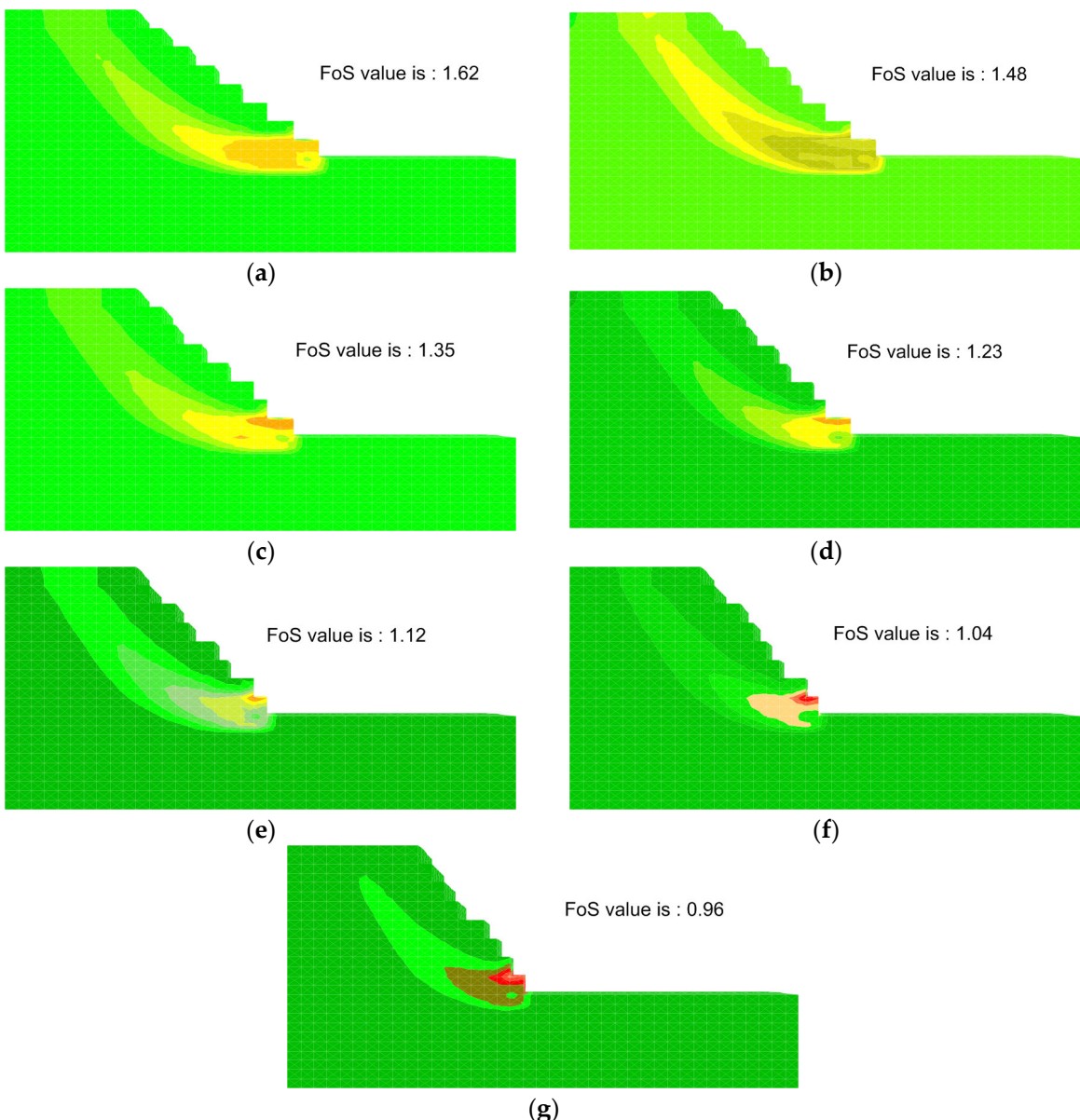

**Figure 10.** Safety factor corresponding to different slop angles when the aging parameter deteriorates by 5%. (**a**) α = 36°; (**b**) α = 38°; (**c**) α = 40°; (**d**) α = 42°; (**e**) α = 45°; (**f**) α = 48°; AND (**g**) α = 51°.

In the same way, the $F_s$ of seven slope forms can be obtained when the aging parameters of slope are deteriorated by 10%, 15%, 20%, and 0%. Considering that the damage of rock is a progressive failure process, and the degradation rate is inconsistent in different periods, the classical Freundlich model is selected to fit the heterogeneous degradation process. The fitting curves are shown in Figure 11.

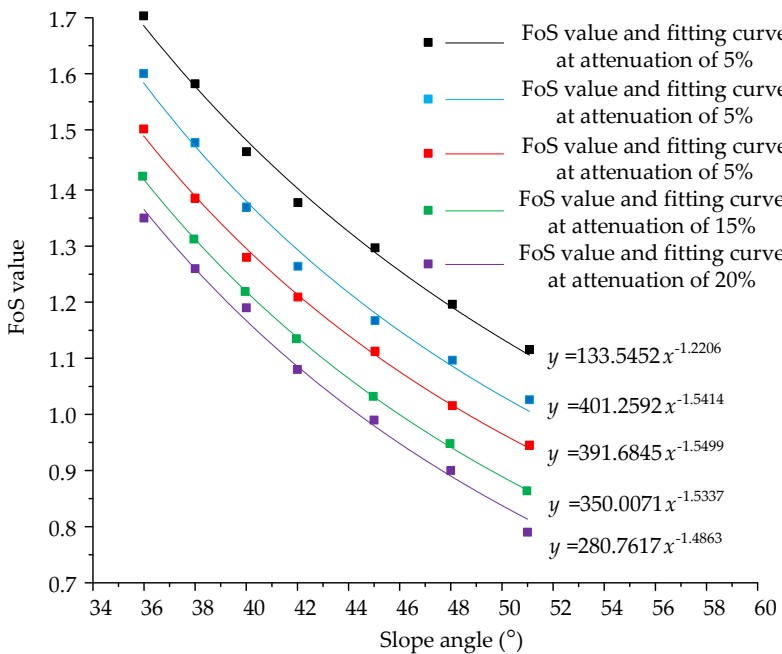

**Figure 11.** Fitting curves of slope angle and safety factors when aging parameters deteriorate by 5%, 10%, 15%, 20%, and 0%.

The corresponding slope angle can be calculated when Fs is 1.3, 1.2, 1.1, and 1.0 by the fitted curve function as shown in Table 4.

**Table 4.** Slope angles corresponding to different reduction degrees and safety factors.

| $F_s$ | ±0% | −5% | −10% | −15% | −20% |
|-------|------|------|------|------|------|
| 1.3 | 44.5 | 41.2 | 39.8 | 38.4 | 37.2 |
| 1.2 | 47.5 | 43.4 | 41.9 | 40.5 | 39.3 |
| 1.1 | 51.0 | 45.9 | 44.3 | 42.8 | 41.6 |
| 1.0 | 55.1 | 48.7 | 47.1 | 45.6 | 44.4 |

It can be seen that the value of slope angle depends on the deterioration degree of slope mechanical parameters. The more serious the deterioration of slope aging parameters, the smaller the slope angle should be designed. On the contrary, when the slope aging parameters are not significantly deteriorated, the degree of slope angle can be appropriately increased. This provides a reference for the adjustment of slope angle. For long-term slopes, due to the long service life, the stability of the slope needs to be guaranteed throughout the service life. Here, the slope parameters of the BP mine are used to carry out the slope timeliness design with a safety factor 1.2. The model is divided into four layers from bottom to top, and the rock and soil properties of each layer are degraded at a growth rate of 5%. At the same time, the angle of each layer is designed according to the slope angle calculated in Table 4. The established aging slope model is shown in Figure 12. It can be seen that from the topsoil layer to the lowermost coal seam, the attenuation of the rock and soil mass in the vertical direction decreases with the increase of the depth, while the corresponding slope angle gradually increases, showing a convex shape as a whole. The shape of the slope reflects the design concept of an aging slope with a gentle upper and a steep lower.

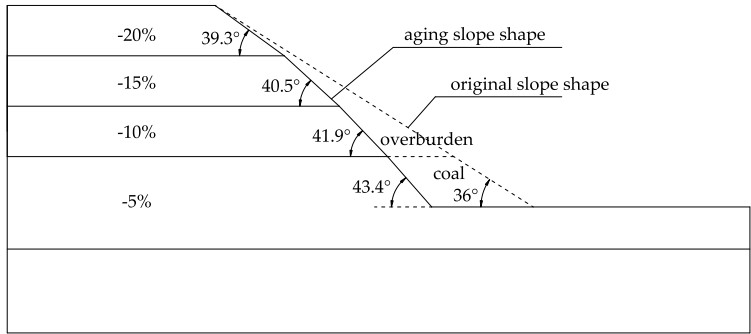

**Figure 12.** Long-term aging slope model.

According to the long-term aging slope model, the slope angle of each layer is obtained by adjusting the bench width of side steps, and the corresponding aging parameters are assigned to each layer. The safety factor is solved by FLAC3D. For the convenience of comparison, the safety factor of slope without attenuation of parameters of each layer is calculated at the same time as shown in Figure 13.

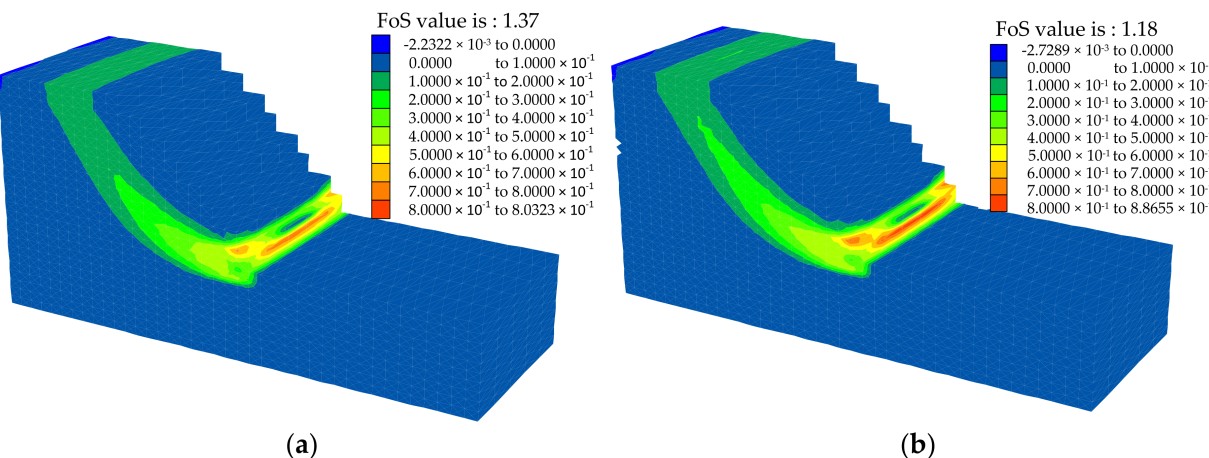

(**a**)                                                    (**b**)

**Figure 13.** Calculation results of long-term aging slope stability. (**a**) Before parameter attenuation; (**b**) after parameter attenuation.

The calculation results show that after the design of the aging slope angle of Fs = 1.2, the safety factor of the slope is reduced from the original 1.70 to 1.37. Further considering the aging difference of the parameters of different rock layers with time, the safety factor of the long-term aging slope is finally 1.18, which is 0.02 lower than the design safety value 1.20, with a decrease of about 1.67% within the acceptable range. Comparing the shear strain increment diagrams before and after the parameter attenuation, it can be seen that the shear strain increment increases after the parameter attenuation, and the width and variation range of the shear strain band increase significantly, indicating that after the excavation of the slope, the properties of the rock and soil mass of the slope show varying degrees of attenuation in the vertical direction with the passage of time, and the plastic penetration zone inside the slope tends to increase gradually. The slope designed considering the time difference can more truly reflect the actual stable state of the slope.

Due to the difference of slope deterioration in the vertical direction, the slope is designed as a convex slope with a gentle upper and a steep lower. The upper boundary is fixed, and the lower steps are retracted as shown in Figure 12. As can be seen, when the overall slope angle is adjusted from 36° to the convex slope with the bottom slope angle of 43.4°, the cross-sectional area of the stripping is 330.31 m², while the cross-sectional area of the recovered coal is 312.59 m². Taking the density of coal as 1.54 t/m³, the stripping ratio of sidewall mining is only 0.69 m³/t. According to the mining speed of 200 m/a

and the mining rate of 95%, 91,500 tons of coal resources can be recovered every year. It can be seen that after the secondary design of the original slope into a long-term aging slope, the resource recovery rate of the mine can be improved, and the economic benefit is very significant.

## 5. Conclusions

Rock masses in cold regions are often damaged by freeze–thaw effects due to the circadian and seasonal climate change. This paper analyzes the instability mechanism of saturated fractured rock slope in open-pit mine under the joint action of repeated freeze–thaw cycles and confining pressure. The sensitivity analysis of crack initial length, adjacent crack spacing, crack inclination angle, buried depth, and crack ratio is performed. The analysis results are applied to the ageing slope design of the BP mine in Xinjiang. Through the design of convex slope shape, 91,500 tons of coal resources are recovered. Aging analysis and design of rock slope is a complex project. Analyzing slope stability in cold regions using numerical simulation can help identify potentially dangerous regions of slopes. In addition, more different crack shapes as well as more loads are also necessary to consider in future work to extend the applicability of aging slope theory. The main conclusions and suggestions are drawn below:

(1) Under the combined action of repeated freeze–thaw cycles and confining pressure, the rock mass of the mine in the seasonally frozen area develops a composite compression-shear expansion at the tip of the crack. The theoretical frost-heave force increases linearly with the increase of mining depth. The rock mass with small shear modulus, small fracture toughness, and large Poisson's ratio is more prone to frost-heave fracture failure. As the inclination angle of the fracture changes from the horizontal to the vertical direction, the theoretical frost-heave force gradually decreases until it tends to be stable, and the change rate is small in the near-horizontal and vertical directions and large in the inclined direction 20–70°.

(2) Taking the BP mine as an example, 35 groups of slope models with different aging strength parameters as well as slope shapes were designed by selecting the internal friction angle, cohesion, and gravity that have a significant impact on the simulation results as attenuation factors. The aging slope angle corresponding to the given safety coefficient was determined by functional fitting of the numerical simulation results. According to the difference in service life at different positions of open-pit slope, the design concept of long-term aging slope is innovatively proposed.

**Author Contributions:** Conceptualization, Z.C.; methodology, Z.C. and W.Z.; software, W.Z.; data curation, F.D. and W.Z.; formal analysis, X.G.; investigation, W.Z. and G.Z.; resources, Z.C.; writing—original draft preparation, X.G.; writing—review and editing, Z.C. and G.Z.; visualization, Z.C.; funding acquisition, Z.C. and W.Z. All authors have read and agreed to the published version of the manuscript.

**Funding:** This research was funded by the Natural Science Foundation of Xinjiang Uygur Autonomous Region (2021D01B34, 2022D01A52), the Young Doctoral Science and Technology Talents of Xinjiang (2020Q080), New Engineering Research and Practice Project of the Ministry of Education (E-KYDZCH20201826), and the Doctoral Initiation Fund of Xinjiang Institute of Engineering (2019BQJ012207).

**Data Availability Statement:** All data supporting the findings in this study are available from the corresponding author on reasonable request.

**Conflicts of Interest:** The authors declare no conflict of interest.

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
