# Peer review of "Aging Stability Analysis of Slope Considering Cumulative Effect of Freeze–Thaw Damage—A Case Study"

_minerals, doi:10.3390/min12050598_

Round 1

Author Response

Cover Letter

Dear Editor and Reviewer 1,

Thank you very much for your comments concerning our manuscript entitled “Aging Stability Analysis of Slope Considering Cumulative Effect of Freeze-thaw Damage” (ID: minerals-1687757). These comments are quite valuable and helpful for us to further improve our paper. Therefore, as you suggested, we have revised our paper and made the according response, as shown below:

Comment 1: Due to the case implementations, please kindly add “case study” to the title of the manuscript.

Response: As suggested by the reviewer, “a case study” has been added.

Comment 2: The concluding remarks of the abstract are not well-written. They are merely the repetition of the objectives and the title of the manuscript. Please provide quantitative findings, model limitations, and verifications in the abstract.

Response: The abstract has been rewritten according to the Reviewer’s suggestion, and the quantitative findings, model limitations, and verifications have been provided.

Comment 3: The necessity & novelty of the manuscript should be presented and stressed in the “Introduction” section.

Response: A paragraph has been added to present the necessity of the manuscript at the end of “Introduction” section.

Comment 4: Provide a literature of the methods developed/applied on coal mine steam or excavation-filling process in the “Introduction” section. The use of a table to demonstrate the advantages-disadvantages of these methods can be useful. Towards the end, mention the superiority & repeat the novelty of your work.

Response: A table to demonstrate the advantages-disadvantages of the mining methods applied on coal mine excavation-backfilling process has been added in page 2.

Comment 5: The methodology section is weakly written. So, my suggestion is to reconstruct it. Use flowcharts &model step-by-step decryptions.

Response: In the methodology section, the description of modeling process and the flowchart of Figure 9 have been added in page 11.

Comment 6: Please add a subsection clearly articulating the main limitations, wider applicability of your methods, and findings in the “Discussion” section. Also, please deepen the “Discussion” section.

Response: Figure 8 has been replaced and a paragraph has been added to present the applicability of the aging slope theory compared with traditional design methods.

Comment 7: What type of slope failures were considered in the rock mass to generate the cracks and which criteria was considered to cover the crack propagation in slope needs to be described.

Response: For deep open-pit mines, the stripping of rock is essentially the horizontal stress unloading while the dumping along the open-pit end wall is vertical stress loading. During the excavation of slope, the lack of effective solid support in the direction of the free surface leads to the loss of the lateral restraint, and a large area of the unloading zone is formed. the rock mass may undergo unloading rebound deformation and instability along the direction of stress loss. At the same time, for the open-pit mine located in the seasonal freezing area, the volume expansion caused by the water phase transformation in the crack will also exert the frost-heave tensile stress in the crack perpendicular to the crack surface. According to the above stress conditions, considering the combined effect of the frost-heave tensile stress and the three-dimensional confining pressure in the crack, the criterion for cracking of fractured rock mass under freeze-thaw condition is determined by applying the principle of stress superposition and the theory of strain energy density factor, and the theoretical frost-heave stress required for cracking is deduced.

Comment 8: The background of the study lacks literature related to uncertainties in crack generation and propagation in solid materials like rock. How did you account for crack tip displacements or kinked crack in rock materials? Please describe.

Response: The background of the study related to uncertainties in crack generation and propagation in solid materials like rock has been added.

After excavation of the open-pit mine slope, stress concentrations may be formed near the crack tip for the internal primary cracks under the unloading stress field consisting of the initial stress (compressive stress) and the unloading stress (tensile stress), causing the cracks to expand unstably. The distribution of such cracks is relatively regular, mainly concentrated near the slope surface in the way of high frequency and short spacing cluster , and approximately parallel to the slope surface. For the convenience of modeling, the cracks are simplified as a series of equal-width and straight-line collinear cracks, so kinked cracks are not considered here. In this paper, the theoretical frost-heave stress required for freeze-thaw crack cracking is derived, and the sensitivity analysis of crack initial length, adjacent crack spacing, crack inclination angle, buried depth, crack ratio is performed. For saturated micro cracks, the theoretical frost heave force required for the micro crack length is small. When the frost heave force generated by the water phase transition in the crack is greater than the theoretical frost heave force, stress concentration will occur at the end of the crack, prompting cracks to grow along the crack direction, which may cause the original crack length be expanded. As the crack length expands, the theoretical frost heave force required for fracture also increases and gradually approaches the actual frost heave force, so the frost heave damage range gradually stabilizes. Then, when the actual frost-heave force does not meet the ultimate tensile stress of rock mass cracking, the crack will not directly propagate, but a progressive damage zone will be formed at the crack tip. Damage accumulation to a certain extent will lead to the initiation and propagation of new cracks.

Comment 9: How have the authors considered circular failure in a jointed rock mass needs to be explained.

Response: In rock mass, if the joints are irregularly distributed, it is impossible to determine the sliding surface through one or more discontinuities or gaps, which may form a shear surface that slips as a curvature or circle.

Comment 10: I noticed that the “Conclusion” section tends to repeat the abstract and the results. The conclusion paragraph should be short, impactful, and direct the reader to this research’s next steps and opportunities.

Response: The “Conclusion” section has been rewritten according to the Reviewer’s comment.

Comment 11: I strongly suggest that the authors review and include the following studies to improve the manuscript.

Response: Thank you very much for your suggested papers which provide a wider context and reasonable ideas and methods for our research. We have referred these papers in our research.

Once again, thank you very much for your comments and suggestions. Looking forward to hearing from you.

Yours sincerely,

Zhiguo Chang

E-mail: changzg@cumt.edu.cn

Reviewer 2 Report

Review on “Aging Stability Analysis of Slope Considering Cumulative Effect of Freeze-thaw Damage”

Based on theoretical analysis and numerical simulation, this manuscript studies the aging stability of slopes under the influence of freezing and thawing in open-pit mines. The results of this manuscript show that the design of slope considering different aging strength parameters can improve the resource recovery rate of mines, and which have certain engineering guiding significance.

This manuscript has the following insignificance:

  1. The quality of English needs improving. It is noted that manuscript needs careful editing by someone with expertise in technical English editing paying particular attention to English grammar, spelling, and sentence structure;
  2. The statement in the conclusion part of the paper is not comprehensive enough, and the goals and results of the study are not clear to the reader;
  3. The major novelty or difficult for the current work need to be formulated clearly, not limit listing reference. Introduction can be enlarged by supplying some recent work;
  4. Figure 10 should show all the fitted curves when the slope aging parameters deteriorate by 0%, 5%, 10%, 15% and 20% respectively for the convenience of readers.
  5. The definition of in the listed formulas is not clear, and there is no corresponding explanation and definition.

Comprehensive Evaluation: I recommend this paper for publication in Minerals after minor revision.

Author Response

Cover Letter

Dear Editor and Reviewer 2,

Thank you very much for your comments concerning our manuscript entitled “Aging Stability Analysis of Slope Considering Cumulative Effect of Freeze-thaw Damage” (ID: minerals-1687757). These comments are quite valuable and helpful for us to further improve our paper. Therefore, as you suggested, we have revised our paper and made the according response, as shown below:

Comment 1: The quality of English needs improving. It is noted that manuscript needs careful editing by someone with expertise in technical English editing paying particular attention to English grammar, spelling, and sentence structure.

Response: We have revised the whole manuscript carefully and tried to avoid any grammar or syntax error. We hope revised manuscript is now improved and meets the requirements.

Comment 2: The statement in the conclusion part of the paper is not comprehensive enough, and the goals and results of the study are not clear to the reader

Response: The “Conclusion” section has been rewritten according to the Reviewer’s comment.

Comment 3: The major novelty or difficult for the current work need to be formulated clearly, not limit listing reference. Introduction can be enlarged by supplying some recent work.

Response: According to the Reviewer’s comment, we have referred 10 new papers in our research to provide a wider context, and a paragraph has been added to present the major novelty of the manuscript at the end of “Introduction” section.

Comment 4: Figure 10 should show all the fitted curves when the slope aging parameters deteriorate by 0%, 5%, 10%, 15% and 20% respectively for the convenience of readers.

Response: Figure 10 has been redrawn which contains the fitting curves of slope angle and safety factors when aging parameters deteriorate by 5%, 10%, 15%, 20% and 0%.

Comment 5: The definition of in the listed formulas is not clear, and there is no corresponding explanation and definition.

Response: We have checked the full text and the uncommented arguments have been annotated.

Once again, thank you very much for your comments and suggestions. Looking forward to hearing from you.

Yours sincerely,

Zhiguo Chang

E-mail: changzg@cumt.edu.cn

Round 2

Reviewer 1 Report

My review comments have been sufficiently addressed and the paper in its present form may be published.